# Design of a Technique for Accelerating the WSN Convergence Process

**DOI:** 10.3390/s23218682

**Published:** 2023-10-24

**Authors:** Jozef Papan, Ivana Bridova, Adam Filipko

**Affiliations:** Department of Information Networks, Faculty of Management Science and Informatics, University of Žilina, 010 26 Žilina, Slovakia; ivana.bridova@fri.uniza.sk (I.B.); filipko@stud.uniza.sk (A.F.)

**Keywords:** fast reroute, OMNeT++, AODV, BFD, WSN, INET

## Abstract

A wireless sensor network (WSN) is a network that monitors the physical environment using small and energy-efficient sensor devices. The wide application of WSNs has caused them to be used in critical applications that require a quick response, even at the cost of higher consumption. In recent years, Fast Reroute (FRR) technology has been developed, which accelerates network recovery after line or node failure. This technology plays an important role in connection recovery and data recovery, which helps speed up detection and redirect traffic. In our work, we created a new modification of the Ad hoc On-Demand Distance Vector (AODV) routing protocol, where we added the fast detection of link failure used in the FRR area. This modification rapidly increased connection recovery time and was tested in the OMNET++ simulation environment. The modification was implemented based on an additional RFC 5880 Bidirectional Forwarding Detection (BFD) module, which speeds up failure detection by sending quick “Hello” messages.

## 1. Introduction

The present time brings demand for the use of Internet of Things (IoT) devices, which results in an increasing load on the transmission networks and devices ensuring transmission in the wireless sensor network (WSN) [1]. A higher load on the network makes it more susceptible to outages, which affects the proper functioning of critical services, ensuring, for example:Security (sensors can collect sensitive data, and it is critical to ensure the authentication and authorization of devices in the network, secure data transfers, and protect against attacks) [2,3,4];Management of the energy level of individual sensors, since WSNs have a limited energy capacity, i.e., monitoring the state of the batteries to ensure constant power supply to the network [5,6];Efficient routing and data transfer are crucial, given limited energy resources and limited bandwidth. Effective routing algorithms minimize energy consumption, maximize the use of available resources, and ensure reliable data transmission [7,8];Fault detection [9,10] and self-repair [11] (a combination of mechanisms such as monitoring, diagnostics, and sensor redundancy ensure that networks are fault-tolerant, highly available, and able to automatically adapt to changing conditions [12]).

Fast network recovery mechanisms are designed to minimize the time required to re-establish a connection. They focus on outage detection and providing an alternate path for data that would otherwise be lost during convergence [13,14].

This work is focused on WSN technology [15], which is used in IoT. Convergence in such a network is important, as it is also used by applications where the reliability of data transmission is critical and battery life is problematic. In WSNs, the key element of network convergence is routing protocols. The choice of a specific protocol depends on various factors, such as application requirements, network size, etc. The most commonly used routing protocols in WSNs are Low-Energy Adaptive Clustering Hierarchy (LEACH) [16], Dynamic Source Routing (DSR) [16,17,18], Sensor Protocols for Information via Negotiation (SPIN) [19], Threshold-Sensitive Energy Efficient sensor Network protocol (TEEN) [20,21], and Ad hoc On-Demand Distance Vector (AODV) and, precisely because of its wide usability and good simulation support, AODV was chosen as the basis for the design and implementation of a new module that will speed up WSN recovery after an outage.

The presented article deals with the acceleration of the WSN convergence of the AODV protocol. The exceptionality of this solution lies in fast network recovery (FRR), which will enable connection continuity in the event of a line or node failure. The AODV protocol allows us to effectively deal with dynamic topological changes in the network. Implementing a new additional module based on RFC 5880 (BFD) is an important element that can speed up error detection, even in complex network environments. The proposed solution can speed up recovery in a wireless sensor network, which is a specific type of network that requires effective solutions for fast recovery. The proposed solution focuses on this area.

The goal of this solution is to create mechanisms that will not only speed up the detection and recovery of the network but also ensure the reliability of the network. This is important in a variety of applications, from commercial networks to critical infrastructures.

So, excellence lies in the combination of these factors, which focus on fast, reliable, and efficient network recovery in different contexts.

As part of our long-term research, we use the OMNeT++ simulator to verify new techniques in the field of FRR, which contains a complete and especially functional library for using the AODV protocol. Although the AODV protocol is quite old, its use is proven by current scientific works [22,23,24,25,26]. However, the principle of this acceleration can also be applied to other WSN protocols.

The first section describes the basics of the WSN, a fundamental principle of the Fast Reroute. The second section is devoted to related works. This section provides a comprehensive overview of recent solutions in the focus area. The third section provides the theoretical basis for solving the problem. The fourth section describes the determination of the research hypothesis. The fifth section proposes a new fast detection add-on to the AODV protocol. The sixth section evaluates the proposed solution and shows the scientific results. The seventh section describes a discussion of the newly designed solution. The last chapter summarizes the proposed work.

## 2. Related Works

This chapter presents what solutions other authors bring to fault tolerance for WSNs.

FRR mechanisms can calculate a backup route before a line or node failure is detected. In order to minimize packet loss during the site convergence process, the backup path is calculated locally on the router [27,28]. 

The ongoing development of FRR mechanisms leads to the creation of two types of FRR solutions—reactive and proactive. Approaches proactively precalculate alternative FRR paths [29,30]. This precalculation will ensure the achievement of a quick change in route to an alternative route.

Proactive IP FRR mechanisms include the Loop-Free Alternate (LFA) [31] and Multiple Routing Configuration (MRC) [32]. The LFA mechanism uses next-hop selection without a loop. MRC uses multiple routing configurations on routers. The source router inserts the configuration number into the IP packet header, and the receiving router selects the correct routing configuration based on this number.

The IP Fast Reroute LFA mechanism has several proposed improved versions, such as directed LFA [33] and Topology-Independent LFA (TI-LFA) [33,34].

Only a few existing FRR solutions, such as LFA and R-LFA, are implemented in current routers. The FRR technology itself was also implemented on programmable switches [35].

MPLS-TE (MultiProtocol Label Switching–Traffic Engineering) was developed for MPLS networks [36].

Alternative distribution trees reconstruct another possible mechanism in the field of FRR [37,38,39]. The most widely used mechanism based on distribution trees is Maximum Redundancy Trees (MRTs) [40,41].

Another option is reactive IP Fast ReRoute mechanisms that do not calculate alternative routes in advance. These include Multicast Repair (M-REP) [42] and Enhanced M-REP [43]. Table 1 shows the evaluation of the properties of individual FRR solutions.

### Problem Areas

Current research in WSNs deals with various problem areas that present new challenges for researchers. Their solution is essential for creating efficient and reliable sensor networks with many applications, such as environmental monitoring, smart cities, etc. Problem areas in WSNs are, for example:Energy efficiency: One of the main problems in WSNs is the limited energy capacity of sensor devices. Research seeks to find ways to minimize energy consumption at the level of sensors, communication protocols, and algorithms to extend network lifetime and enable long-term deployment [44].Data processing: Sensors in WSNs acquire a huge amount of data that need to be processed and interpreted. Research focuses on the development of algorithms for data processing and analysis, including clustering, classification, aggregation, and anomaly detection [45].Security: Considering sensitive data and critical applications, research deals with the development of mechanisms to ensure authentication, encryption, data integrity, and protection against attacks [46].Network topology: WSNs often have a dynamic topology, as sensors can be mobile or out of service. The research deals with the design of algorithms for topology management, change detection, network reconfiguration, and ensuring reliable connection in a dynamic environment [47,48].Efficient routing: It is an important aspect of WSNs. Since sensors are often spread over large areas, it is necessary to find optimal paths for data transmission from one point to another. Research in this area deals with the design of efficient routing protocols that consider energy constraints, network topological properties, FRR mechanisms, and others [8].

As part of our research, we are focusing on the last-mentioned research area, effective direction, and we established the following subproblem areas:Fast network recovery (Fast Reroute, FRR): This area deals with the development of techniques and protocols that enable fast connection recovery in the event of a link or node failure in the network. The goal is to minimize the impact of outages on operations and ensure that the network restores its functionality as quickly as possible.Fault detection and traffic rerouting: This area focuses on identifying outages and faults in the network and then rerouting traffic to alternative paths. The goal is to minimize the impact of a failure on the connection and ensure continuous operation of the network.Ad hoc On-demand Distance Vector protocol (AODV): The AODV routing protocol is widely used in ad hoc wireless networks. The goal is to verify the effectiveness and efficiency of this protocol in the context of rapid network recovery and traffic rerouting after an outage.OMNET++ simulation environment: The OMNET++ simulation environment aims to test the designed add-on module and evaluate its impact on the speed of fault detection, packet loss prevention, and network recovery process.New add-on module based on RFC 5880 (BFD): This area deals with the design and implementation of a new module in the AODV routing protocol, which is based on RFC 5880 (Bidirectional Forwarding Detection—BFD) standards. The goal is to use BFD to speed up fault detection and improve network recovery after an outage.

Overall, in this work, we investigate fast network recovery (FRR) techniques and their application to ad hoc wireless sensor networks. The goal is to create efficient and reliable mechanisms for solving outages, detecting faults, and quickly restoring network operations.

## 3. Theoretical Principles Used in Research

### 3.1. Fast Reroute (FRR)

FRR mechanisms serve to ensure fast and reliable transmission in a network, minimize the impact of an outage on operation, and thus improve the availability of services. The principle of operation of the FRR mechanism is presented in [49].

There are two different approaches to Fast Reroute in IP networks. The speed at which these accesses can be performed depends on the network conditions, namely the network topology, available resources, and the type of error or event.

A local repair is usually faster than a global repair, because it is limited to the area around the fault location. In the event of a local repair, backup routes or route redirection within the local circuit will be used to work around the error. This approach is faster because it does not require changing the overall network topology.

On the other hand, a global fix would involve changing the entire network topology, which can be time-consuming and complex. This approach is suitable if the fault is very serious and affects a large part of the network.

The exact speed ratio between local and global patching depends on the specific circumstances and technologies used in the network. In some cases, an in-place repair can be hundreds of times faster, while in other cases, it can only be a few times faster. It is a matter of network design and specific implementation details.

An example of comparing these two types of repairs from the point of view of speed is shown in the following Figure 1.

Global repair (Global Fast Reroute) in computer networks is performed through various techniques and protocols that ensure fast bypassing of errors or failures in the network at the overall topological level. These techniques and protocols may include the following [50,51]:

Multiprotocol Label Switching (MPLS): MPLS is a technology that enables efficient routing of data flow in the network based on labels instead of traditional routing based on IP addresses. In the event of a failure, an alternative MPLS path (fast error bypass) can be created via another route [52,53].

Interior Gateway Protocol (IGP) Fast Reroute: Many interior gateways in a network, such as Open Shortest Path First (OSPF) or Intermediate System to Intermediate System (IS-IS), can support fast error workarounds through various mechanisms such as congestion connections or microcircuits that allow immediate redirection of the data flow [54].

Border Gateway Protocol (BGP) Multipath: In the case of multipath BGP configurations, routers in the network can have multiple alternative paths. In the event of an error, these alternate paths can be used to quickly redirect the data flow [55].

Segment Routing: This is an innovative routing technique that allows network paths to be defined using segments or stages, without the need for complex routing tables. Segment Routing can be used to implement fast workaround for network errors [52].

The exact techniques and protocols used for global patching may vary depending on the specific network implementation and availability and speed requirements. The goal is to ensure that in the event of network errors or failures, data are quickly and efficiently rerouted along alternative routes to minimize service disruption and data loss.

The main advantage of local repair is its speed and efficiency. By using the shortest possible alternative path, traffic can be quickly rerouted without significant network disruption. This is particularly important when high levels of availability are required, such as for networks where critical application or service data pass. The time required for local repair is an important factor by which FRR mechanisms are compared [10,11,12].

When analyzing FRR mechanisms, it is necessary to consider the following:The level of computational complexity of the algorithm (precomputing);Effectiveness of network protection (repair coverage);Options for protection against line or router failure;Fault detection time and communication recovery time;Percentage success rate of saved packets;Multicast technology support.

Currently, there is no ideal FRR mechanism. The goal of the developed mechanisms is to optimize the performance and reliability of the network and thus minimize the negative impact of failures on the operation itself [56,57].

### 3.2. Fast Recovery of the WSN

The WSN is a network focused on monitoring the physical environment using small and energy-efficient sensor devices. These sensors collect data on various variables such as temperature, humidity, lighting, or movement and transmit them via wireless communication. A WSN is often used in applications such as environmental monitoring, smart homes, industrial control, and so on.

In WSNs [58], routing takes place by sending data between sensor devices in the network and the final destination. Routing in a WSN has some specific characteristics compared with traditional networks.

The basic flow of routing in a WSN is described below:

Topology distribution: At the beginning of routing, sensor devices in the network inform each other of their presence and availability. This can be performed by periodically sending and receiving signals between sensors or by using dedicated routers.

Choosing a router: In some WSNs, special sensors are designated that perform the function of routers. These routers have the task of receiving, processing, and transmitting data from other sensor devices. The choice of a router can be determined based on various factors, such as energy, distance from the target, availability of neighboring sensors, and so on.

Creating a routing table: Each sensor device in the network creates a routing table that contains information about neighboring devices and the best path to the destination. Routing tables can be updated based on a change in topology or network status.

Message routing: When a sensor device wants to send a message to a destination, it uses a routing table to decide the optimal path. The message is then passed from one sensor device to another in the network until it reaches its destination.

Handling missing messages: Due to the limited energy and resources of sensor devices, messages may be lost, or data transmission errors may occur. Therefore, techniques for handling missing messages, such as retransmission, buffers, patch codes, and so on, are often used in WSNs.

The choice of the routing protocol is important, since the nondelivery of a data packet to the destination node may occur, for example, due to a packet collision, a channel or node failure, radio interference, or the movement of nodes, when the node has moved out of range of another node.

The most used reactive routing protocol is AODV. Such a protocol does not create any alternate paths, so there is only one path, which in the event of an outage causes nodes to start dropping data packets until a new path is created. This outage can cause unwanted problems if the system transmits data requiring real-time communication. How fast recovery options can be added to this protocol is described in the subsections below.

### 3.3. Bidirectional Forwarding Detection (BFD)

The BFD mechanism is designed for quick detection of line outages in the network. The mechanism works independently of the routing protocol used in the given network, thanks to which BFD can detect a fault in the network in a uniform time. Using this mechanism makes the most sense for protocols where outage detection time is long or outage detection is nonexistent. The messages of the BFD mechanism are called “Hello messages”. The principle of BDF is a three-way mechanism, where a pair of nodes periodically send each other BFD messages. If one of the nodes stops receiving messages, it marks this connection as broken. BFD can operate in two modes:Asynchronous mode: Based on the constant periodic sending of control messages, and in case of their nondelivery, the connection is cancelled;Demand mode: Messages are sent only in case of a request from another node, when the status of the neighboring node is verified [59,60].

Both modes can also work in Echo mode. If the Echo function is enabled, a stream of BFD packets is sent to the neighbor, which sends these packets back unchanged. All that is needed is signaling to which session these BFD packets belong. If a certain number of BFD packets are not forwarded back, the session is considered unavailable (down) [60]. If this function fulfills the role of detection, the interval between the periodic sending of BFD packets can be longer (in the case of Asynchronous mode) or completely disabled (in the case of Polling mode) [59].

Pure Asynchronous mode has the advantage of needing half the amount of BFD packets to achieve a specific detection time compared with Echo. It is also suitable for deployments where the Echo function cannot be supported for certain reasons.

The echo function has the advantage that it only really tests the path to the neighboring router. Such an approach can reduce round trip jitter and thus speed up the detection time. The Echo function can be turned on for each direction individually.

Query mode is useful in situations where the overhead of periodic logs is difficult, such as a system with many BFD sessions. It is also useful when the Echo function is switched on symmetrically. The Query mode has a disadvantage in the detection time because it depends on the heuristics of the implementation, and the BFD protocol itself has no idea about it. The Polling mode cannot be deployed if the round-trip time is greater than the required detection time [59].

A BFD session can be initialized in two ways:Active role—a system with such a role must establish a BFD session by sending a BFD packet to a neighboring node. It does not matter whether the BFD packet of the given session has already been received.Passive role—a system with this role must not start sending BFD packets for a given session unless it receives a BFD packet from a neighboring system. BFD creates a session only when it receives a BFD packet.

To successfully establish a BFD session, at least one system between which the BFD session is being established must have an active role. Once the session is established, the systems begin sending slow, periodic BFD control packets. After three-step verification, the session becomes fully enabled, “UP”. In this state, it is possible to turn on the query mode or the additional Echo function.

### 3.4. Ad Hoc On-Demand Distance Vector Protocol (AODV)

AODV is a reactive routing protocol in which links are established between nodes only when needed, and the protocol initiates a search whenever a source node (S) needs to reach an unknown network [61]. The line is kept alive as long as resource S needs it, or as long as the multicast node group exists. Links between nodes are always loop-free thanks to sequence numbers. A sequence number is created by the destination node for each piece of route information that is sent from the source node. When it is possible to choose between two paths, the path with the higher sequence number is selected. Nodes maintain the next hop information in their routing table. Four types of Hello messages are used in this protocol, Route Requset (RREQ), Route Reply (RREP), Route Error (RERR) [62,63].

Hello messages are used to monitor the state of the link to a known neighbor and are periodically broadcast to the neighbor. If the neighbor is not able to receive the Hello message, the line between them is evaluated as nonfunctional. If the source S is looking for a route to an unknown destination (D), it sends a so-called RREQ. In each node between S and D, information about the link to node S is stored. In case this new node is found, or the node already has a link to resource S, it starts generating RREP. The RREP is sent unicast hop-by-hop back along the route to the source S. After the source S receives the RREP, it has a functional route to the destination D. If S receives multiple RREP responses, i.e., multiple routes exit between S and D, S chooses the shortest one according to the hop count.

When data flow between S and D, each node stores time data for each route. If a route has not been used for a certain period, the node cannot know whether it is operational and removes that route from its routing table. Subsequently, it sends RERR in the direction of node S. On the line to S, each node deletes the record of the currently nonfunctional route. After S receives the RERR, it is unable to send data to D and restarts the route search by sending an RREQ if necessary. The principle is shown in Figure 2 and Figure 3.

### 3.5. AdHoc On-Demand Distance Vector–Backup Routing (AODV-BR)

The AODV protocol does not use alternative paths. Consequently, when a route is disconnected, the nodes of the broken route drop the data packets because there is no alternative route to the destination available until a new route is established. Therefore, mechanisms are being sought to eliminate this problem [64].

Backup Routing is a mechanism for creating an alternative path that extends the functionality of the original AODV protocol. Alternative routes are created at the phase when the response to the route search (RREP) is returned. Once received, RREP slightly modifies the functionality of the AODV protocol. Nodes in the network use the nature of wireless communication when they intercept packets transmitted by neighboring nodes. They will use the packets captured in this way to create an alternative path. Since RREP messages are sent unicast but still using radio communication, neighboring nodes capture these messages, and if this packet is not directed to it, it records the neighbor from which it captured this message in its alternative routing table. This node will be closer to the destination node as the next hop. If a node captures multiple RREP messages for the same route, the node is in the radio range of multiple primary route nodes. In this case, it selects the best route in terms of capture time and stores it in its alternative routing table. After the primary path is established, nodes containing entries in the alternate routing table are an active part of the network. The alternative routes together with the primary route form a herringbone structure (Figure 4).

After an outage, a re-search for the path is started by the source node S. This search is faster compared with the search using classic AODV, because the neighboring nodes keep the path to the destination D in the alternative routing table. Since the convergence time is shorter, there are also smaller packet losses [65,66].

### 3.6. Implicit Backup Routing–AODV (IBR-AODV)

Like the previous mechanism, this one also extends the functionality of the AODV routing protocol. This mechanism works on the principle of overhearing transmitted data by neighboring nodes. If a node supporting such a mechanism is within the range of three nodes of the primary route, it creates an alternative routing table after interception and stores the “overheard” data in a buffer. The following shows the principle of creating a backup node. The primary route is S -> R1 -> R2 -> D (Figure 5), and node R3 is a backup node because it is within range of nodes S, R1, and R2, from which it listens for data communication [67,68].

In the case when a failure occurs on the primary route, for example, one of the nodes moves away or fails completely, the backup node detects this and initiates the start of route recovery. The backup node sends a Route Change (RC) packet to the source node or to a node trying to transfer data. This node updates its routing table and sends an Acknowledgment (Ack) of receiving the RC to the backup node. After receiving the Ack, the backup node loads the data from the alternative routing table into the main routing table and deletes the alternative. Subsequently, the backup node sends the data to the next node, which updates its routing table. After this step, the data routing is fully moved to the backup route. Such a case is shown in Figure 6, when node R1 fails and the primary route S -> R1 -> R2 -> D changes to the backup S -> R3 -> R2 -> D after recovery.

### 3.7. Hybrid Routing

Hybrid routing protocols have the advantages of both proactive and reactive routing protocols. They are trying to balance the latency and the complexity of the overhead in the network. The proactive part of hybrid routing oversees routing in an area where nodes are known and available. The reactive part takes care of sending requests to areas outside known nodes. The entire operation of hybrid routing is based on areas. All nodes are partitioned into areas that use reactive and proactive features for route maintenance and discovery. In general, hybrid routing reduces overhead and performs better under dynamic changes in the network. An example of a hybrid protocol is the Zone Routing Protocol (ZRP).

With the ZRP, most of the communication occurs between nodes located close to each other. In principle, however, each node has its own zone. The size of the zone is assigned by the so-called Joan radius. This number indicates how many hops from the given node belong to its area. Each node has many overlapping regions, and each region can be of different sizes. However, there are the following three components of ZRP [69,70]:IntraZone Routing Protocol (IARP): It is used for communication in an area with known nodes (proactive component). A route to a destination in the local area can be created from IARP routing tables, which are stored in the memory of individual nodes.InterZone Routing Protocol (IERP): It is used for communication outside the area of known nodes (reactive component) [71].Bordercast Resolution Protocol (BRP): It is used to increase the efficiency of communication outside the area of known nodes [72].

### 3.8. Simulation Environment for Wireless Networks

There are a few simulation environments that are used to simulate wireless sensor networks (WSNs). These environments provide tools for modeling and simulating the behavior of sensor devices, communication protocols, and various network scenarios. Table 2 shows an overview of the most frequently used simulators for the WSN.

In the next work, the OMNeT++ simulator, which uses the C++ programming language and is component-based, with modular and open architecture of the simulation environment and with the support of a graphical user interface, will be used for simulation in the WSN. It is a universal simulator capable of simulating any system composed of interacting devices. Thanks to several extensions, OMNet++ is suitable for wireless mobile network simulations.

## 4. Determination of Research Hypothesis (RH)

For the purposes of our research, a research hypothesis (RH) was established. The development of the solution is divided in five steps:Step 1: Review of current fast network recovery techniques and convergence principle;Step 2: Analyzing existing mechanisms for rapid network recovery;Step 3: The design and implementation of own fast recovery mechanism (a new additional module based on RFC 5880 (BFD) was implemented into the OADV protocol);Step 4: Verification in the OMNET++ simulation environment;Step 5: Evaluation of the achieved results.

The effort is to obtain a solution that minimizes the impact of outages on the operation in the network as the convergence time of the network is accelerated. In the end, it will be confirmed or refuted (RH).

## 5. Proposal—Design and Implementation of a Quick Recovery Mechanism (Step 3)

Acceleration of the AODV protocol is realized by the implementation of BFD. The testing could not be performed in real conditions, so the available simulation environments that can be used for the simulation of WSNswere examined, and the OMNeT++ simulator was subsequently selected. The BFD protocol was implemented in the form of Asynchronous mode according to RFC 5880.

Figure 7 shows the integration of BFS fast failure detection into the hierarchy of the AODV protocol. In view of the fact that each verification of a specific link through BFD means an excessive load and consumption of electricity, the BFD session is activated only on links that are critical from the point of view of the infrastructure and are determined by the administrator.

The BFD protocol is a failure detection protocol that is often used in routing protocols for fast path failure detection. The AODV protocol is designed without direct support for BFD; therefore, for the implementation of BFD in AODV, the following was necessary:Get to know the BFD and AODV protocols in detail.Identify places in AODV where BFD could be integrated.Create an independent BFD module that ensures communication between neighboring nodes and performs outage detection functions.Integration of the BFD module into AODV—the created BFD module must be integrated into the existing AODV code, that is, adding BFD functions at appropriate AODV times and ensuring proper coordination between BFD and AODV operations.Testing and debugging the integrated BFD in AODV and verifying the proper functioning of outage detection.

Because modifying an existing routing protocol can affect its performance, reliability, and correctness, thorough validation and testing, is important to ensure proper network functionality. In the following sections, a short description of the actual implementation of the BFD module in the OMNeT++ INET library is presented.

### 5.1. BFD Packet

First, it was necessary to create a control packet. A BFD packet is a message exchanged between neighboring nodes to signal their availability and detect outages. The BFD packet contains the following fields:Version: This field identifies the version of the BFD protocol. The version currently in use is BFD version 1.Diagnostic: The diagnostic field is used to provide diagnostic information about the state of the connection. It can contain values such as “No Diagnostic”, “Control Detection Time Expired”, and others.State: This field represents the current state of the connection between neighboring nodes. It can contain values such as “Admin Down”, “Down”, “Init”, “Up”, and “Unknown”.Poll: This field is used to poll a neighboring node for a response. When set to 1, it means that the node requests a response from the neighboring node.Final: This field is used to end the BFD session. When set to 1, it indicates that the node is trying to terminate the connection.Control Plane Independent: This field indicates whether the BFD communication is independent of the control plane (for example, independent of OSPF or BGP). When this field is set to 1, BFD communication is independent of the control plane.Detection Time: This field represents the time interval between sending BFD packets. Specifies how often BFD packets should be broadcast between neighboring nodes.Desired Min TX Interval: This field specifies the minimum interval between sending BFD packets from the node that is the source of BFD messages.Required Min RX Interval: This field specifies the minimum interval between receiving BFD packets on the node that is the destination of BFD messages.

BFD control packets have a mandatory section and an optional authentication section. Only the mandatory part was implemented in the presented work; the authentication part will be the subject of further research.

When writing the C++ code, the option of defining the packet in the MSG file was used. The task of the OMNeT++ message compiler, opp_msgc or opp_msgtool, is to translate the definitions into C++ classes that are created from the C++ code model. The message compiler is invoked for .msg files automatically as part of the build process. A BfdControlPacket.msg file was created in which the BfdControlPacket class was defined with its necessary fields. In order for the fields of this class to be subsequently used when creating and sending a packet, it was necessary to define that it would extend FieldsChunk. Thanks to this extension, the packet data structure will be built on another set of data structures called chunk-y. Chunks provide several alternatives for representing parts of data. In the created file, the namespace for the packet, i.e., inet and BFD, were additionally defined. To represent the state of the BFD session, the BfdState enum was created, which defines the necessary numerical values as constants. For the other parameters of the BFD control packet, the most suitable data types according to their sizes were used. Figure 8 shows the BfdControlPacket.msg file.

### 5.2. BFD Session

The next step is to implement the BFD session. To avoid creating unnecessary BFD sessions between all devices, how and when these sessions are created on the active path was designed.

The active path is the defined path traveled by the ping request or its response. The creation of an AODV routing path was used when the node also creates a BFD session when returning or creating an RREP. A node that establishes a BFD session in this way is a node with an active role. After creating a session in the BFD list of sessions, a timer is also started to send a message to the neighboring node for which the session is created. The time after which this message is sent was defined to the default value of the sending interval. If the message was sent immediately after the session was established, the messages began to collide with the RREP and RREQ messages that were still traveling through the network. After a message is received by a neighboring node, a session is automatically created on it as well, and thus it becomes a node with a passive role (see Figure 9).

The createBfdSession() function is used to create a session, which at the very beginning makes the use of this function available with the Enter_Method() macro in the AODV functions of the routing protocol. This is followed by the initialization of the BFD session parameters according to RFC 5880 and the return of the created session (see Figure 10).

### 5.3. BFD Reports

After obtaining the correct functionality in the case of failure with correct termination, it was necessary to redefine the handleMessageWhenUp() function. This function serves as a crossroad for received messages. If it is a self-message (a message sent to itself—timers function), it starts to find out for which session (the timer for sending a BFD message or the timer for outage detection) has expired. If a match is found, a function is called, either to send a message based on the unique session number or to handle the outage. If it is not an own message, the received message is processed by the socket (Figure 11).

In the event that the timer for sending regular BFD messages expires, the function sendBfdMessage() is called (Figure 12). The parameter of this function is the local discriminant to know for which session the next BFD packet needs to be sent. At the beginning, the function creates a new BFD control packet with the given discriminant. It then searches the list of sessions, where it finds the given session and sends the created packet. If RemoteMinRxInterval is zero or if demand mode is enabled, according to RFC 5880 [59], a BFD packet must not be sent periodically. Therefore, the necessary conditions are found here. Additionally, a recalculation of the sending interval is added before scheduling the next timer. In the RFC, it is written that this interval is equal to the higher value of the compared values. This value has been adjusted to equal the lower value. When setting a higher value, a poll sequence would have to be invoked for the proper functionality of the mechanism when DesiredMinTxInterval or RequiredMinRxInterval is changed. Setting a smaller value does not cause the timers to be missynchronized, and therefore the session does not break up. The poll sequence has not yet been implemented. At the same time, when the sending interval was changed, the jitter had to be adjusted as well. After adjusting the sending interval and its jitter, another timer was scheduled. At the end of the function, there is an auxiliary extract of the BFD list of sessions.

As mentioned before, a BFD packet is created before sending a message. The createBfdPacket() function is responsible for creating the packet. This function creates a BFD control packet based on the local discriminant parameter and the session assigned to it according to RFC 5880 [59] (Figure 13).

Once created, the packet itself is sent in the sendBfdMessage() function by calling the sendBFDPacket() function. In this function, it is checked at the beginning whether a message with TTL set to 0 is not being sent. In the case of a different number, it continues further, where the name of the class of the packet and the created packet are set, which comes as an attribute and is attached to the end of the new packet. Subsequently, the interface, TTL, address, and port of the destination to which the packet should be sent via the socket are set (Figure 14).

### 5.4. Processing of the Packet

Processing of the packet is performed in the processPacket() function. In it, the source address from which the received packet originates is determined, then the BFD control packet is extracted and both of these values are sent as parameters to the next function handleBfdMessage(). Finally, after completing the handling of the received packet, the packet itself is deleted (Figure 15).

In the handleBfdMessage() function, the correct values of the received BFD control packet are verified at the beginning according to RFC 5880. If any condition is met, an error message is displayed, and the packet is immediately discarded. If no condition for dropping the packet is met, the packet is received and searched for in the BFD list of the session according to the source address. If the session with the received source address is not already in the letter, a new session is created and added. Subsequently, in this session, other correct settings of the received packet are verified again, while the packet is again discarded if it fulfills any condition. After verification, a correctly configured packet is processed in such a way that the following hold:If regular sending of BFD control packets is not scheduled, it will be scheduled.The RemoteDiscr of the session is set according to the received value.The RemoteState of the session is set according to the received value.The RemoteDemandMode of the session is set according to the received value.The RemoteMinRxInterval of the session is set according to the received value.If the received packet has the Required Min Echo RX Interval set to 0, the sending of Echo packets must be enabled (not yet implemented).The poll sequence is used if it is set in the received packet (not yet implemented).The detection time is updated according to the received values and the outage detection timer is scheduled.If the session state is administratively down (AsminDown), the received packet is dropped.If the session status of the received packet is administratively disabled (AsminDown), the session status will be set to disabled (Down), if it is not already, and the local diagnostics will be set to 3.If the session status is other than administratively disabled (AsminDown), the session status is set according to the following status machine (Figure 16).

If the session status after evaluating the machine is ON (UP) and the neighboring device also has a session in the UP status, the interval between sending regular BFD packets can be adjusted. Here, only one common rapidTransmissionInterval was implemented, which speeds up the sending of BFD packets. However, each session can have its own pace of sending and receiving packets. This is an option for further development of the mentioned implementation.If the BFD mechanism of the remote device is in demand mode and both devices have a session in the UP state, the local system must stop the regular sending of BFD packets. Demand mode was not implemented in this case because it requires separate detection of the outage. This work is focused on the possibility of detection using the BFD mechanism, and in this case, the demand mode would not help.

In this section, the basic part of the BFD implementation into the AODV protocol is described. In the next section, the testing and evaluation of the proposed solution is rebuilt.

## 6. Evaluation

To test the proposed solution in the form of the implementation of the BFD module (indicated in chapter 4) into AODV, 2 test scenarios were created, the veracity of the RH determined in part III was verified by simulation in the OMNET++ simulation environment.

### 6.1. Scenario 1—Simple Outage

The mechanism was first tested on a simulation with a simple AODV life cycle (Figure 17), namely on the critical node intermediateNodeA and the node that sends the ping request to the sender. The BFD default interval was set to 1 second and the accelerated interval to 100 milliseconds.

Simulation result without BFD mechanism:


--------------------------------------------------------



sent: 177 received: 176 loss rate (%): 0.564972



round-trip min/avg/max (ms): 1.90644/3.85699/336.706



stddev (ms): 25.2391 variance: 0.000637012



--------------------------------------------------------


The result of the simulation with the BFD mechanism on the nodes sender and intermediateNodeA:


-------------------------------------------------- ------



sent: 177 received: 177 loss rate (%): 0



round-trip min/avg/max (ms): 1.90644/2.03401/13.4117



stddev (ms): 1.14656 variance: 1.31461e-06



-------------------------------------------------- ------


The results show that the implemented BFD mechanism managed to save the previously “lost” ping and also significantly reduce the maximum packet travel time. This maximum time is assigned from the detailed listing of the event log to the first broadcast ping that had to wait for the connection to be established. After the outage, it was found that previously, the lost ping had to wait for a shorter time, namely 11.94927 ms.

### 6.2. Scenario 2—Complex Outage

After desimulating the mechanism on a simple simulation, more complex simulations with more nodes continued. First, simulations were performed without node mobility with a guest node failure [16]. After the outage, the simulation continued in the same way as before. And thus the shortest path was found initially through the host node 16. When the host node 0 detected an outage, it subsequently found the shortest path through the host node 19 and the host node 7, after which ping responses were also returned (Figure 18).

Simulation result without BFD mechanism:


-------------------------------------------------- ------



sent: 177 received: 176 loss rate (%): 0.564972



round-trip min/avg/max (ms): 1.90723/55.2861/3811



stddev (ms): 384.227 variance: 0.147631



-------------------------------------------------- ------


The result of the simulation with the BFD mechanism on host (0) and host (16) nodes:


-------------------------------------------------- ------



sent: 177 received: 177 loss rate (%): 0



round-trip min/avg/max (ms): 1.90723/7.37104/813.213



stddev (ms): 60.9186 variance: 0.00371107



-------------------------------------------------- ------


It is clear from the result that the applied BFD mechanism worked similarly to the simulation with a simple life cycle of the AODV routing protocol. It was possible to save the previously “lost” packet and significantly reduce the maximum packet route time. Further simulations took place on the same scenario but with the BFD mechanism turned on at all network nodes. In this case, however, the backup path was created through node host (10) and node host (4) (Figure 19).

Result with the BFD mechanism enabled on all nodes:


-------------------------------------------------- ------



sent: 177 received: 169 loss rate (%): 4.51977



round-trip min/avg/max (ms): 1.90723/31.5021/2462.21



stddev (ms): 219.911 variance: 0.048361



-------------------------------------------------- ------


From the analysis of the event dump, it was found that the sessions after the outage occurred not only on the path host [0] <-> host [10] <-> host [4] <-> host [1] but also between the node host (1) and nodes host (6) and host (7) and between node host (10) and node host (6) (Figure 20).

As seen in Figure 20, sessions were established on two possible routes. After detailed analysis, it was found that the first route created was host (0) <-> host (10) <-> host (6) <-> host (1). However, on the route between nodes host (10) and host (6), there was a loss of three consecutive BFD control packets, so the BFD mechanism detected the outage, thus initiating the finding of a new path. It was found just through the second shown route through the nodes host (0) <-> host (10) <-> host (4) <-> host (1).

## 7. Discussion

To verify the results of the presented simulations, additional tests were created, which served to verify the results and contributed to the confirmation of the hypothesis established in Section 3.

In Scenario 1, a simple AODV life cycle simulation was presented. Five different tests verified this simulation, the results of which are presented in Table 3. All simulations were run for 180 simulation seconds. They differ in the number of nodes on which the BFD mechanism is running and the settings of the detectMult (detection multiplier) and rapidTransmissionInterval (accelerated BFD packet sending interval) parameters. Packet loss represents the percentage value for how many “ping” requests did not receive a response. The average packet travel time is the average time of all successfully received pings. This time is measured from the origin of the ping request to the successful delivery of its response. Recovery time is the time required to restore the connection after an outage. The time is measured from the detection of the outage to the restoration of the connection when the sending node is again allowed to send the ping request. The following tests were performed:−Test 1 is a simulation without the BFD mechanism.−Test 2 is a simulation with the BFD mechanism not activated on any node.−Test 3 is a simulation with the BFD mechanism activated on the sender and intermediateNodeA nodes.−Test 4 is a simulation with the BFD mechanism activated on the nodes sender, intermediateNodeA, and intermediateNodeB.−Test 5 is a simulation with the BFD mechanism activated on all nodes.−Each scenario was repeated 20 times to confirm and validate the result.

Our implemented BFD mechanism reduced the average packet travel time just by applying it without actively enabling it on any node. This application caused some shift in the event scheduler, where it prevented the RREP from being lost during the initial creation of the AODV path. It can also be seen from the results that our implemented mechanism worked without problems during the expected outage, where it was applied only to the sender and intermediateNodeA nodes. These nodes are the center of the outage, as the sender node broadcasts traffic, and the intermediateNodeA becomes the outage itself. The other scenarios initially resulted in packet losses, which we managed to solve by adjusting the parameters and thus achieving the required zero loss rate, reducing the average packet travel time and reducing the network recovery time.

We also achieved similar results when applying our mechanism to a simulation with a larger number of nodes, specifically to a simulation without mobility with a host node failure [16]. The achieved results are shown in Table 4. To test the proposed Scenario 2, five different tests were carried out:−Test 6 as a simulation without the BFD mechanism.−Test 7 as a simulation with the BFD mechanism not activated on any node.−Test 8 as a simulation with the BFD mechanism activated on host [0] and host [16].−Test 9 as a simulation with the BFD mechanism activated on all nodes.−Test 10 as a simulation with the BFD mechanism activated on all nodes except the host node [1].

The design of the mechanism was that it would compensate for increased energy consumption by increased fault detection for a critical node or critical part of the network. According to the results achieved, we managed to successfully implement this proposal. However, if we use the mechanism for the entire network, it is advisable to adjust parameters such as detectMult and rapidTransmissionInterval in the BFD mechanism. We also recommend turning on the BFD mechanism so that the outage is detected only in one direction, thus preventing double propagation of the RERR message. The Echo function could help to solve this problem.

AODV + BFD takes advantage of fast failure detection using specialized fast BFD packets. With these properties, we can detect failures much faster than classic AODV and other protocols. Table 5 shows the comparison of classic AODV average Round Trip Time (RTT) to AODV with integrated BFD protocol.

According to the results shown in the table, AODV with BFD is faster by approximately 48% in comparison with basic AODV.

Another positive aspect of the integration of the accelerated link drop detection module into AODV is that there is only minimal packet loss. In critical infrastructures, this aspect is very important, as the minimum amount of data are lost.

## 8. Conclusions

At the conclusion of our study on fast routing in the event of a link or node failure, we can state that these techniques have an impact on ensuring the reliability and efficiency of communication networks. Our research efforts focused on testing the AODV routing protocol in the OMNET++ simulation environment and implementing a new add-on module based on RFC 5880 (BFD). This module was able to speed up the detection of network failures through simple “Hello” messages, resulting in faster and more efficient recovery of wireless sensor networks.

Our effort contributes to the development of effective and reliable mechanisms for solving outages, detecting faults, and quickly restoring network operations, which has the potential to increase the performance and reliability of communication infrastructures. This work paves the way for further research efforts and innovations in the field of high-speed routing and network security to better prepare networks to meet demanding challenges and ensure continuous connectivity.

The method proposed by us to speed up the convergence process is suitable for use in risky parts of the network, where it can ensure increased availability and reliability of individual nodes. Its disadvantage is the waste of the energy resources of the devices.

The future direction of research in the field of FRR will be towards the security of networks and their resistance to attacks, as the growing threat of cyberattacks brings the need to develop techniques that are able to effectively solve network outages caused by attacks but at the same time ensure the preservation of integrated security mechanisms.

Overall, the future direction of research in the field of FRR and network security will be towards even more reliable, faster, and smarter solutions that will be able to face the demanding challenges of modern communication networks.

## Figures and Tables

**Figure 1 sensors-23-08682-f001:**
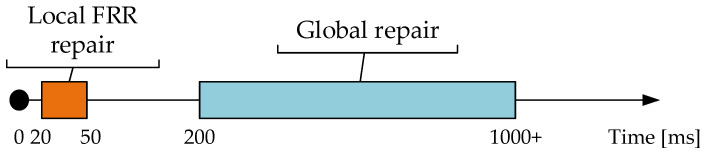
Comparison of local and global correction.

**Figure 2 sensors-23-08682-f002:**
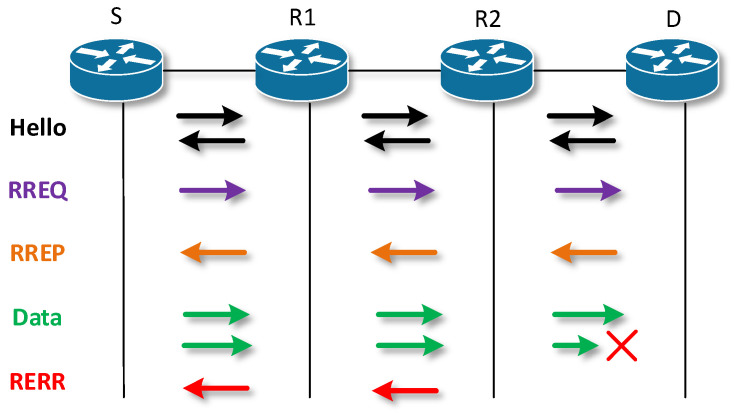
Sending messages in the AODV protocol.

**Figure 3 sensors-23-08682-f003:**
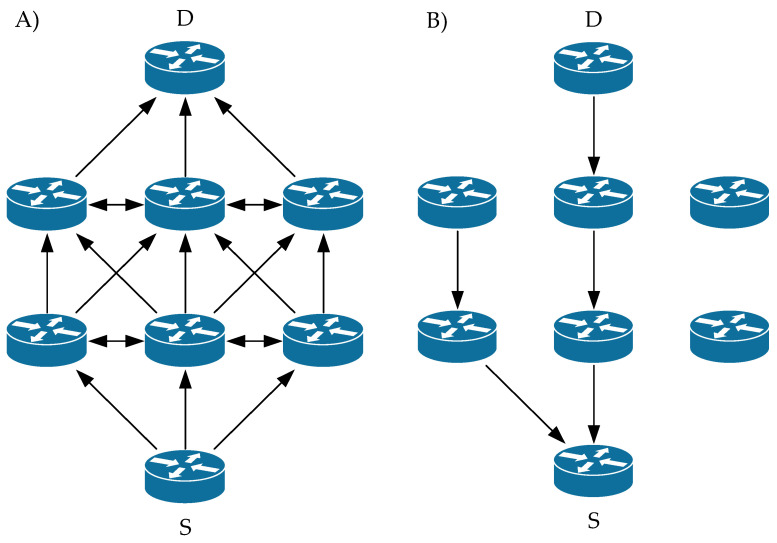
Propagation of RREQ (**A**) and RREP route (**B**).

**Figure 4 sensors-23-08682-f004:**
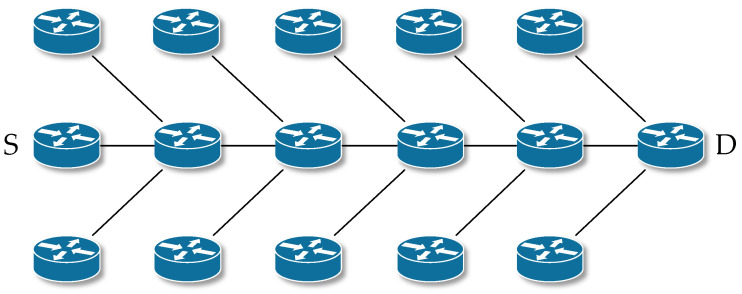
Structure of AODV-BR network.

**Figure 5 sensors-23-08682-f005:**
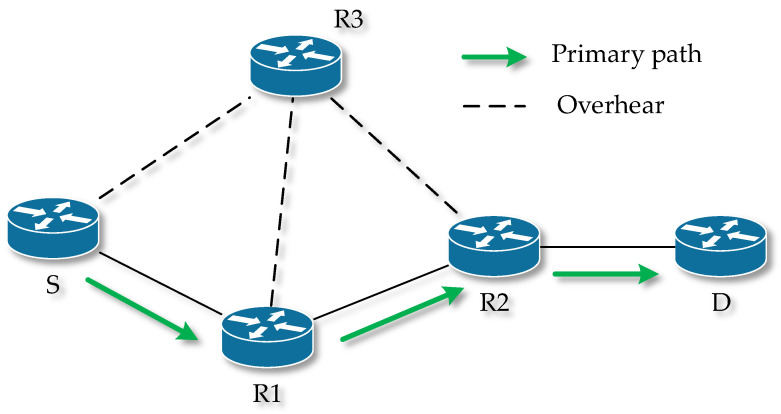
The principle of creating a backup node IBR-AODV.

**Figure 6 sensors-23-08682-f006:**
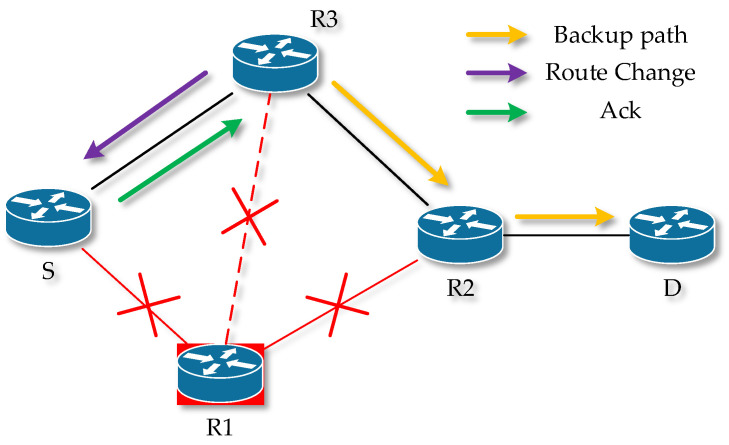
Redirection in IBR-AODV.

**Figure 7 sensors-23-08682-f007:**
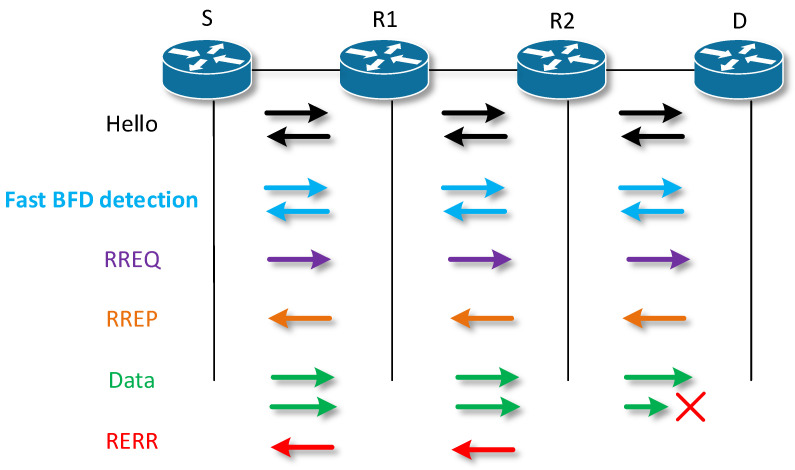
Integration of fast BFD Hello messages.

**Figure 8 sensors-23-08682-f008:**
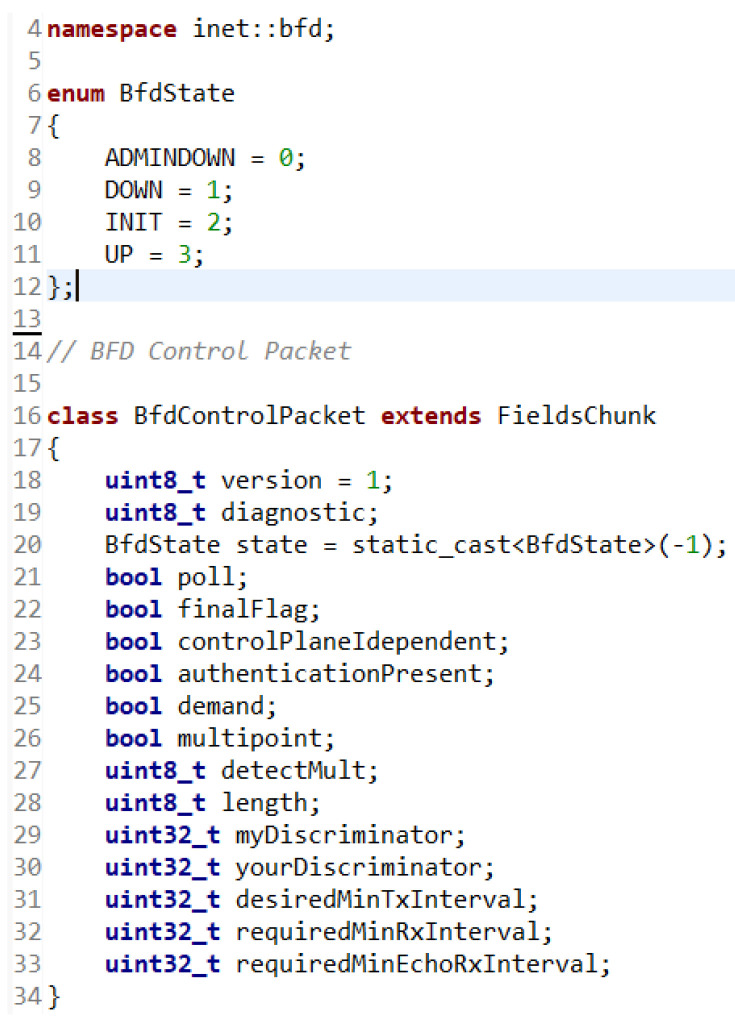
BfdControlPacket.msg file.

**Figure 9 sensors-23-08682-f009:**
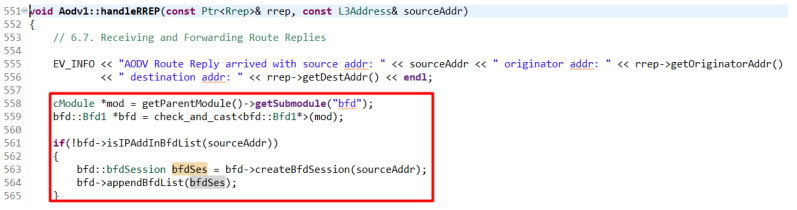
Establishing a BFD session on receipt of an RREP.

**Figure 10 sensors-23-08682-f010:**
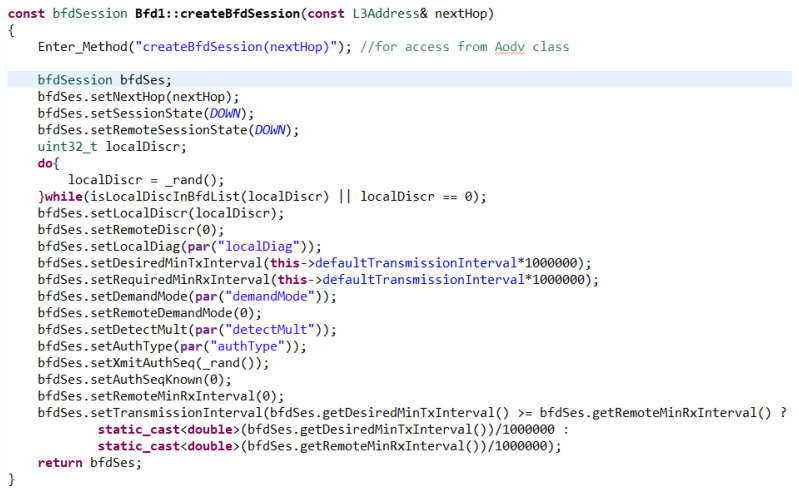
Function to create BFD session.

**Figure 11 sensors-23-08682-f011:**
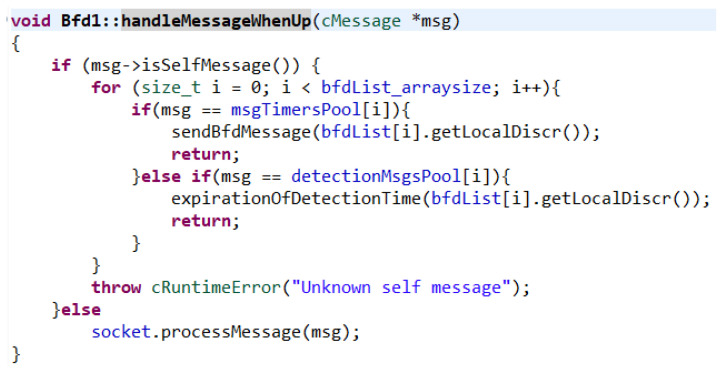
HandleMessageWhenUp() function.

**Figure 12 sensors-23-08682-f012:**
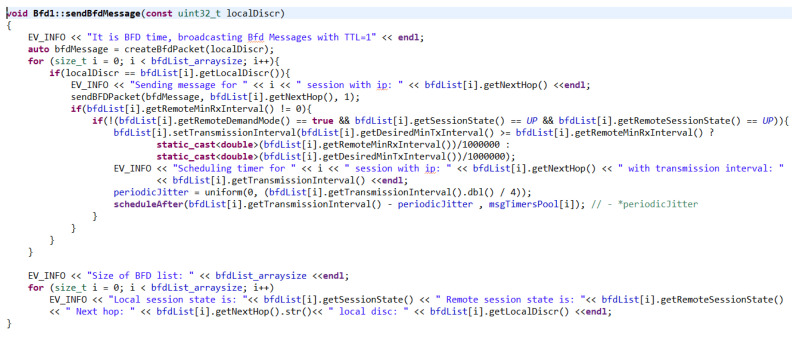
SendBfdMessage() function.

**Figure 13 sensors-23-08682-f013:**
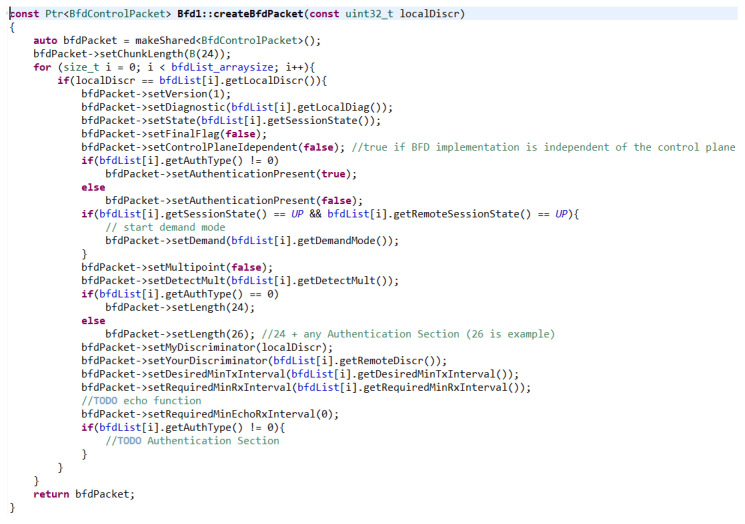
Function createBfdPacket().

**Figure 14 sensors-23-08682-f014:**
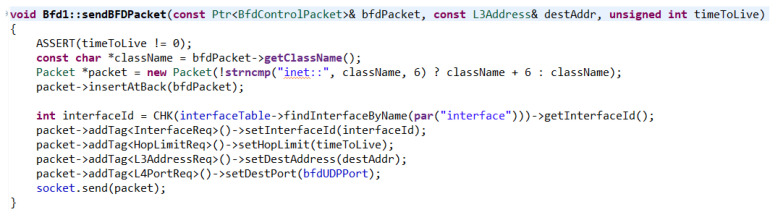
SendBfdMessage() function.

**Figure 15 sensors-23-08682-f015:**
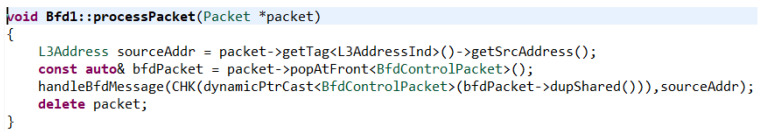
ProcessPacket() function.

**Figure 16 sensors-23-08682-f016:**
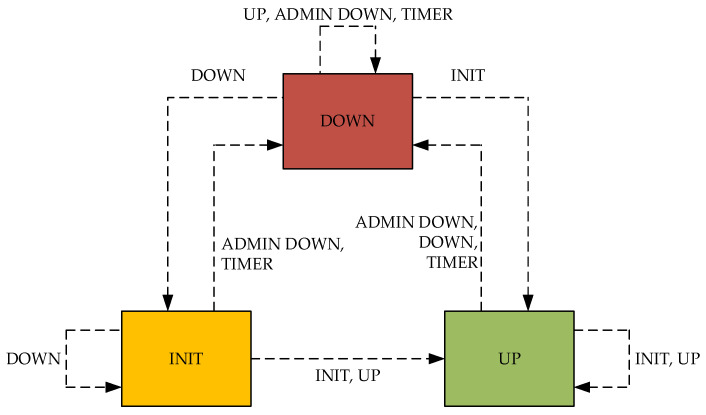
BFD state machine.

**Figure 17 sensors-23-08682-f017:**
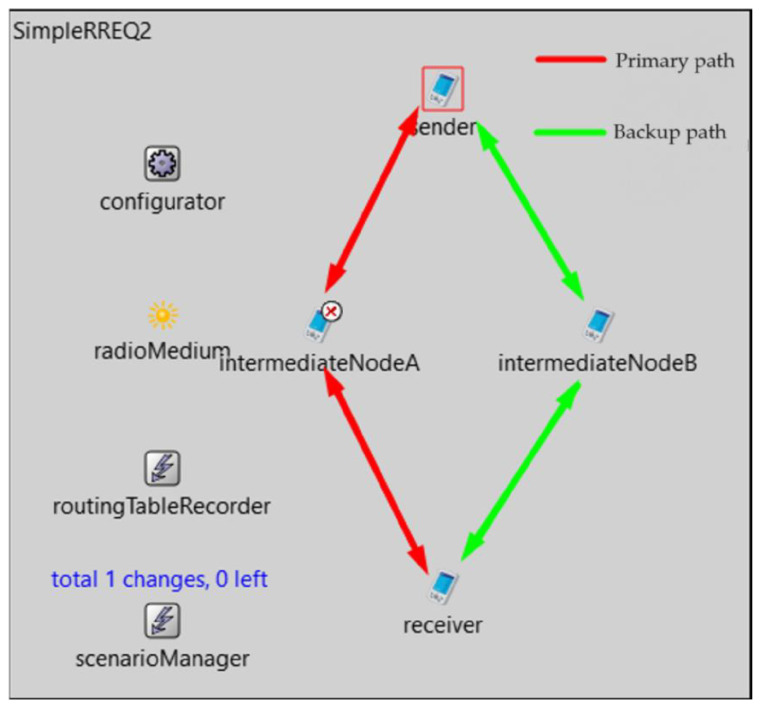
Simulation with a simple AODV life cycle.

**Figure 18 sensors-23-08682-f018:**
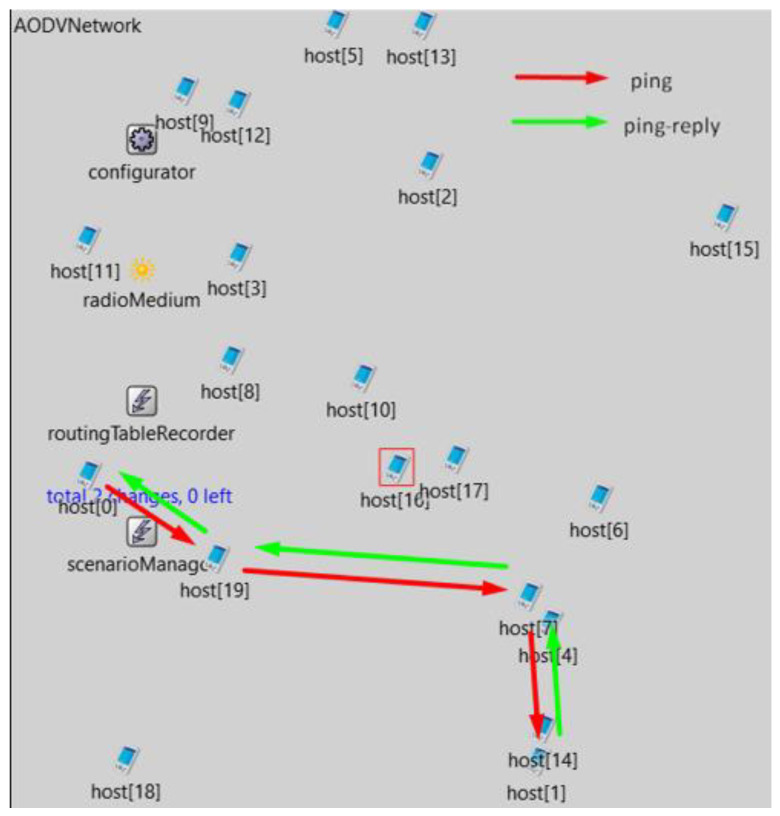
Simulation with BFD enabled on nodes host [0] and host.

**Figure 19 sensors-23-08682-f019:**
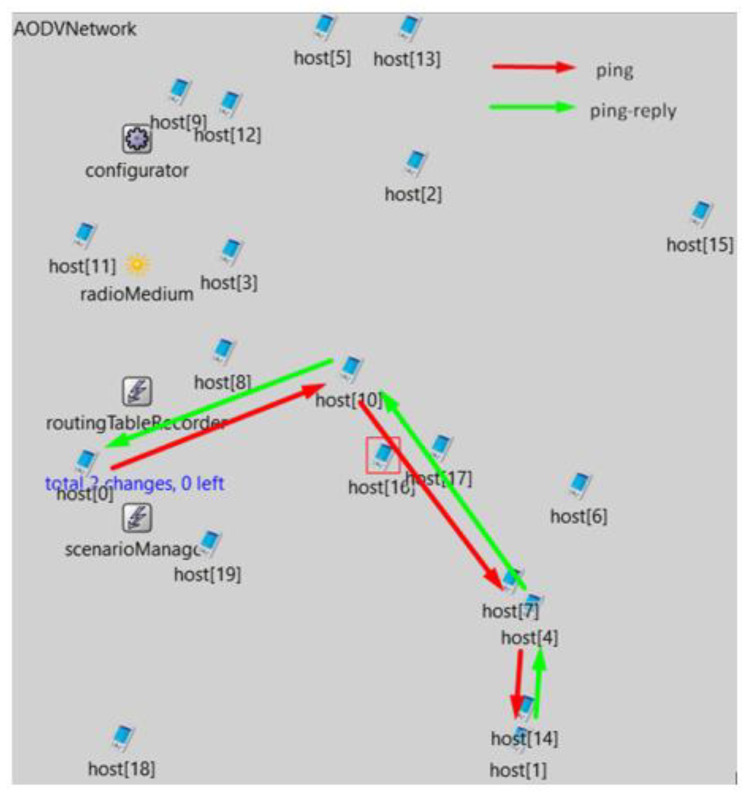
Simulation with BFD enabled on all nodes.

**Figure 20 sensors-23-08682-f020:**
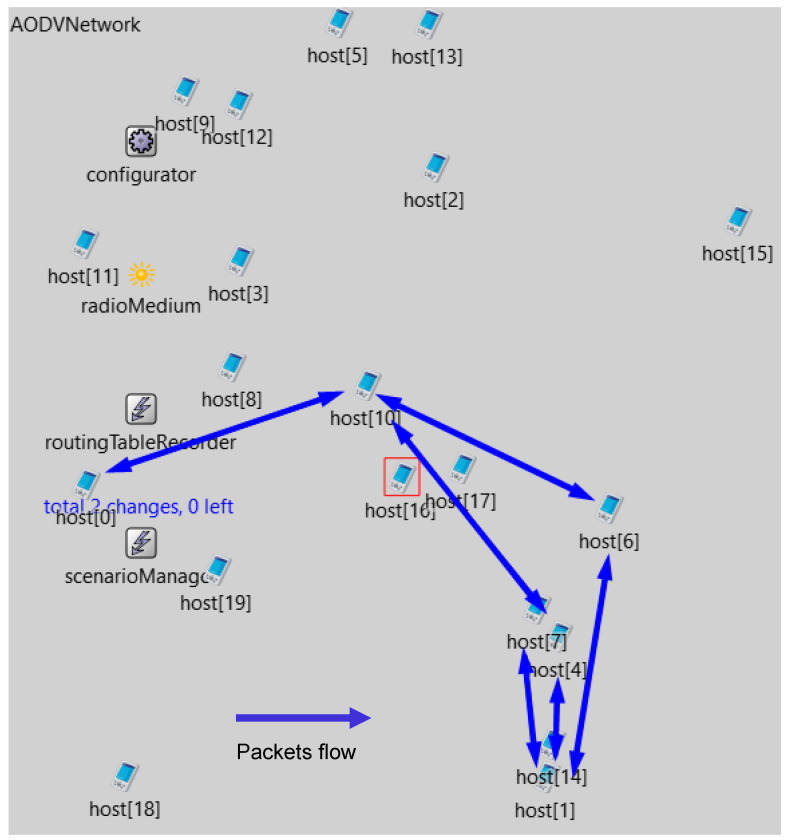
Created BFD sessions.

**Table 1 sensors-23-08682-t001:** Comparison of existing FRR mechanisms.

	B-REP	EM-REP	MRC	MRT	LFA	R-LFA	D-LFA
100% Repair Coverage	Yes	Yes	Yes	Yes	No	No	Yes
Custom Alternative Path	Yes	No	Yes	No	No	No	No
Precomputing	Yes	No	Yes	Yes	Yes	Yes	Yes
Packet Modification	Yes	Yes	Yes	Yes	No	Yes	Yes
Link-State Dependency	Yes	No	Yes	Yes	No	Yes	Yes

**Table 2 sensors-23-08682-t002:** Simulation tools.

Name	License
ns-2 [73]	Open source
GloMoSim [74]	Open source
OPNet [75]	Commercial
QualNet [76]	Commercial
OMNeT++ [77,78]	Open source
COOJA	Open source
J-Sim [79]	Open source
SWANS [80]	Open source
TOSSIM [81]	Open source
ns-3	Open source
MiXim [82]	Open source

**Table 3 sensors-23-08682-t003:** Simulation results of a simple AODV life cycle.

The Name of the Scenario	Detection Multiplier	Accelerated Interval (s)	Packet Loss Rate (%)	Average Packet Travel Time (ms)	Recovery Time (s)
Test 1	3	0.1	0.564972	3.85699	0.90747
Test 2	3	0.1	0.564972	2.03286	0.90971
Test 3	3	0.1	0	2.03401	0.66698
Test 4	3	0.1	1.12994	2.03663	0.66698
Test 4	2	0.1	0	2.04288	0.76905
Test 5	3	0.1	1.69492	4.43788	0.92814
Test 5	2	0.1	0.564972	4.84417	1.03019
Test 5	3	0.15	0	4.99224	0.87140

**Table 4 sensors-23-08682-t004:** Results of simulations without mobility with failure of the host node.

The Name of the Scenario	Detection Multiplier	Accelerated Interval (s)	Packet Loss Rate (%)	Average Packet Travel Time (ms)	Recovery Time (s)
Test 6	3	0.1	0.564972	55.2861	2.23644
Test 7	3	0.1	0.564972	4.69619	1.23513
Test 8	3	0.1	0	7.37104	1.23338
Test 9	3	0.1	4.51977	31.5021	2.86021
Test 9	3	0.15	4.51977	18.3285	2.83879
Test 10	3	0.15	0	18.7073	2.09002

**Table 5 sensors-23-08682-t005:** RTT comparison of AODV without BFD to AODV with BFD.

	AODV	AODV + BFD
Average RTT (ms)	3.85699	2.03401

## Data Availability

The data presented in this study are available on request from the corresponding author. The data are not publicly available due to privacy reasons.

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
