# Peer review of "Design of a Technique for Accelerating the WSN Convergence Process"

_sensors, 2023, doi:10.3390/s23218682_

Round 1

Reviewer 1 Report

1,The abstract should include the background, proposed methodology, and experimental results.

2,At the end of the introduction, the organization of subsequent chapters should be indicated

3,There are two subsections 5 in the text.

4,For the fast recovery mechanism designed in Section IV, draw a flowchart and describe it briefly

5,In related work chapters, present solutions proposed by other authors in the area of fault tolerance in WSN networks. It is more beneficial to show a summary and comparison of the individual solutions.

6,It is recommended to add a table of abbreviations

7,It would be useful to provide more insights into the strengths and potential challenges of the proposed module.

Minor editing of English language required

Author Response

Dear reviewer, please see the file.

Reviewer 2 Report

The paper deals with the design of a new technique to accelerate the convergence process of the WSN network, but it is not well structured: 9 pages of the 20 in total are related to the state of the art divided into Introduction and Related works and only a few pages (considering the number and size of the Figures included) contain the most significant part of the work. I suggest the Authors strongly revise the document by reducing the state of the art and focusing more on the innovation.

The work presented consists of the implementation of a new algorithm and its testing using a simulation, but only 5 tests are shown, why do the Authors decide to perform so few tests instead of hundreds of tests to show a scientifically significant statistic? The proposed method seems to be effective, but the results are very few to convince about the reliability of the methods in real conditions, if we consider an application in a WSN network with hundreds of nodes. I also suggest to better quantify and discuss the waste of energy resources of the devices, as reported on line 698.

Some minor remarks:

1) please use the full name of the algorithm followed by the acronym in parenthesis and not vice versa.

2) line 123: I suggest adding "is described below"

3) please delete line 279; lines 376-392; line 436; line 458; line 503.

Minor editing of English language required

Author Response

Dear reviewer, please see the file.

Reviewer 3 Report

Dear Authors,

Remove the extras * in authors names (line 4).

Fix the emails and extra semi-commas (lines 5-7).

Acronyms should be first shown in extend mode (line 11).

The Abstract should present to the readers some results obtained with this work (after line 17).

Some keywords should be studied more to be used. For example, WSN appears in academic results less than Wireless Sensors Network. So, I suggest using the extended mode and applying the same criteria for the other keywords.

The wrong way to declare an acronym (line 21).

Typo in line 23 (extra dot).

The references should be [2-4] instead of [2],[4] (line 28).

The references should be [5,6] instead of [5],[6] (line 31).

The subsection name is Fast ReRoute (FRR), but at the beginning of the Abstract is declared Fast Reroute (FRR). Thus, keep the same standard for the whole work (line 53).

When a Figure is cited, it is unnecessary to use bold (line 72).

The caption of Figure 2 and Figure 3 needs to be corrected (lines 219 and 223). Please maintain the same standard with the whole work.

Why Section 3 is so short?

Figures 7, 8, 9, 10, 11, 12, 13, 14 should be used as code or pseudo algorithm, never as image.

Please improve the quality of Figures 17, 18, 19.

In general, the authors should re-read the manuscript, focusing on maintaining the standard that the MDPI indicates in this URL: https://www.mdpi.com/journal/sensors/instructions. It is very important to do this.

Author Response

Dear reviewer, please see the file.

Reviewer 4 Report

Design of a technique for accelerating the WSN network convergence process

Few specific facts about this manuscript is given below:

1. The AODV routing protocol has been tested in the OMNET++ simulation environment, how disturbances, irregularities, or variations in signal conditions are considered in this simulation environment?

2. The study only considers the AODV routing protocol and introduces a new module to it based on the RFC 5880. The versatility of the proposed solution across different protocols isn't addressed in the provided description.

3. In WSNs, sensor nodes are usually battery-powered, and thus, power consumption is a critical factor. However, it isn't mentioned whether the additional overhead of the new add-on module, which speeds up fault detection by sending "hello" messages, significantly affects power consumption. If it does, it could limit the applicability of the solution in energy-constrained wireless sensor networks.

4. The described study does not mention anything about how this protocol might scale with an increased chart of nodes in the WSNs. As networks scale up, efficient fault detection and repair become more challenging. It isn't clear how the proposed solution would perform under high load or with a significant number of nodes.

5. It's not specified which performance metrics were used in assessing the utility of the add-on module. Metrics like end-to-end delay, packet delivery ratio, routing load, and throughput can give a holistic view of how the change affected the overall network performance.

6. The solution appears to focus on dealing with single-point outages. Multiple simultaneous faults might occur. How this solution handles multiple faults or cascading faults is not addressed.

Minor editing of English language required

Author Response

Dear reviewer, please see the file.

Round 2

Reviewer 2 Report

Dear Authors, thank you for your reply. I appreciate the efforts to improve the article, but it is not yet well structured, no changes have been made to reduce the state of the art (Introduction + Related works) and even in this version only a few pages (considering the number and size of figures included) contain the most significant part of the work. The number of tests is still 5 and this does not provide a scientifically meaningful statistic to demonstrate the effectiveness of the algorithm and no discussion of increasing the number of nodes is included in the new version. Isuggest to revise again the manuscript before publication.

Author Response

Dear reviewer, please see the file.

Reviewer 3 Report

Dear Authors,

Well done.

Best regards.

Author Response

Thank you for reviewing the article.

Reviewer 4 Report

Authors have addressed  almost all the concerns so the manuscript may be considered for the publications.

Only one suggestion, authors are requested for thorough proof reading.

Author Response

Thank you for reviewing the article, we have improved the grammar.